# TOWARDS SELF-ADAPTABLE ROBOTS: FROM PROGRAMMING TO TRAINING MACHINES

**Víctor Mayoral, Risto Kojcev, Nora Etxezarreta,**
**Alejandro Hernández and Irati Zamalloa**
Erle Robotics
Venta de la Estrella Kalea, 6
01006 Vitoria-Gasteiz, Araba, Spain
`{victor, risto, nora, alex, irati}@erlerobotics.com`

## ABSTRACT

We argue that hardware modularity plays a key role in the convergence of Robotics and Artificial Intelligence (AI). We introduce a new approach for building robots that leads to more adaptable and capable machines. We present the concept of a self-adaptable robot that makes use of hardware modularity and AI techniques to reduce the effort and time required to be built. We demonstrate in simulation and with a real robot how, rather than programming, training produces behaviors in the robot that generalize fast and produce robust outputs in the presence of noise. In particular, we advocate for mammals.

## 1 INTRODUCTION

Today's landscape of robotics is still dominated by vertical integration where single vendors develop the final product leading to slow advances, application specific robots, expensive products and customer lock-in. The *traditional* approach for building robots has empowered this reality. It consists of a cascaded process similar to the one described in Figure 1 a). This procedure is composed of different steps that go from the purchase of robot components, to the deployment of the final robot for a given task. Once executed, it produces a certain output, and any modification in the robot will demand a big engineering effort to maintain the same result. In Figure 1 a), the *critical section* for the *traditional* approach has been highlighted. This path marks the section in the process where each individual change on any of the contained blocks will demand a re-execution of all the following steps inside the critical section. Such a limitation leads to a time and resource-consuming approach for building robots. Still, this process remains being the most popular in industry and leads to robots that lack of flexibility and reconfigurability.

Opposed to this traditional approach for building robots, as introduced by Mayoral et al. (2017), modular robots promise interoperability and ease of re-purposing. Figure 1 b) illustrates the *modular* approach. When followed, the integration effort is removed and the critical section reduced significantly. However, although the process of building robots, and particularly, the integration of new robot modules[1] is simplified, the task of programming robots remains cumbersome. New modules, although interoperate, need to be introduced in the logic of the system manually. This implies that for each module addition or modification, a complete review of the logic that governs the behavior of such robot will need to happen. In other words, the adaptation capabilities of these systems are still limited.

Figure 1 c) illustrates the *Modular And Self-Adaptable* (MASA) approach for building robots. This approach radically changes the robot building process in which, rather than programming, modular robots train themselves for a pre-defined task. By continuously integrating the information from its modules, based on an information model such as the one described by Zamalloa et al. (2018), a robot is able to adapt automatically when new modules are added. This approach reduces both the human development effort and time significantly. The process of building a robot gets simplified to defining a task, adding robot hardware modules and letting the robot train until it accomplishes the assigned task.

---

[1]The term module is used to refer to an interoperable component. Refer to Appendix D for more details.

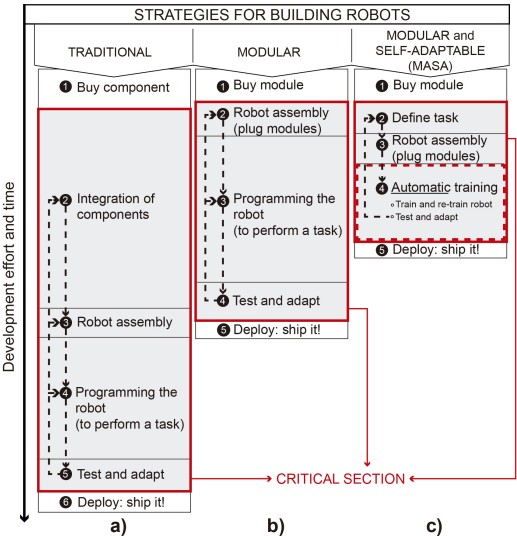

Figure 1: Depicts three different strategies for building robots: a) the traditional approach, b) a modular approach where interoperable modules can be used seamlessly to extend the robot and c) the Modular And Self-Adaptable (MASA) approach where modules, besides interoperating, are detected and configured automatically in the robot for its use.

This paper tackles the problem of how to build self adaptive robots more effectively. Robots that are easy to configure and re-purpose. We present the concept of a self-adaptable robot that makes use of modularity and Artificial Intelligence (AI) techniques to reduce the effort and time required to be built. We demonstrate how rather than programming, training produces behaviors in the robot that generalize fast and produce robust outputs, even in the presence of noise. We prove that training the robot at faster trajectory execution times in simulation preserves the same accuracy when transferred to the real robot. We compare the trajectories produced by the same robot built using the strategies in Figure 1 a) (*traditional* approach) and Figure 1 c) (*MASA* approach). We discuss the results and show how *MASA* leads to self-adaptable robots that could disrupt the future of how these machines will be built and configured.

The remainder of this article is organized as follows: section 2 introduces previous work. Section 3 provides a discussion on the preliminary results obtained. Appendix A includes insight about our bio-inspired approach when defining *MASA*. Appendix B describes the robot used for the experiments. Appendix C presents the setup used for the evaluation of both the *traditional* approach and the *MASA* approach. Appendix D contains definitions for relevant terms used through the article.

## 2 PREVIOUS WORK

The topic of achieving AI through building robots was early discussed by Brooks (1986). In the technical report, the author criticizes the *traditional* approach for building intelligence embedded in robots and argues for a shift to a process-based model , with a decomposition on behaviors[2] as the organizational principle. Ultimately, the author claims that such decomposition leads to improvements on redundancy, speed and extensibility. Inspired by Brooks, we adopt an incremental construction attitude and present a method for building robots that corresponds with Figure 1 c) and we name the *MASA* approach.

The relationship between traditional academic robotics and traditional AI was analyzed by Brooks (1991), where the author concluded that new approaches were garnering interest, yet much work continued in the traditional style. This situation remains a fact. The clearest indicator is that, currently, the Robot Operating System (ROS) Quigley et al. (2009), the *de facto* framework for robot application development, presents many primitives for the traditional approach and few for newer

---

[2]By 'behaviors' we imply what the original paper describes as 'task achieving behaviors'.

paths. More recent articles, such as the work by Rajan & Saffiotti (2017), discuss an integrated approach for AI and Robotics. The authors state that the fields of AI and Robotics drifted apart due to the fundamentally different backgrounds of its practitioners, and present an integrated landscape for Robotics and AI with a clear disembodiment in terms of the robotics challenges. We argue that such approach can not capture the embodied circumstances of robotics. We argue that an integrated path for AI and Robotics demands a strong consideration of the process of building and configuring robots. In particular, we claim that modularity plays a key role in the convergence of AI and Robotics.

## 3 DISCUSSION AND PRELIMINARY RESULTS

Figure 1 c) presents the *MASA* approach for building robots. Its critical section, much smaller than other approaches', has been highlighted in red. Similar to what Brooks (1986) proposed, *MASA* presents a mechanism to incrementally build intelligence for a given task. In the same report, Brooks argues that roboticists[3] typically assume static environments however real-world scenarios involve dynamism. We argue that within these dynamic changes, robots are subject to errors in their measurements, and ultimately, related to their imperfect sensing capabilities.

We present an experiment that aims to shed some light into the relevance of this new approach for building robots meant for real-world scenarios (subject to noise and errors) and on the spot testing. We compare the results obtained by the *traditional* approach (details about the setup available in appendix C.1) and the *MASA* one (details available in appendix C.2) on a given task. The task of the robot is to reach a given point in the workspace. The setup consists of a robot with 3 Degrees-of-Freedom (DoF) in a SCARA configuration (details explained in appendix B). The robot is built and configured by following the approaches of Figures 1 a) and c). The configuration of each robot follows from its building process and is either programmed or trained. Simulation is used to accelerate the process of experimentation applying, when appropriate, *faster than realtime* techniques as introduced by Kojcev et al. (2018). In a first experiment, the robot is built and programmed using traditional control theory mechanisms and is subject to Gaussian noise ($N(0, \sigma)$) introduced on each of its joints to simulate the imperfect sensing capabilities. In a second experiment, a modular robot is trained with motor joint observations that pass through a similar Gaussian filter introducing noise on each iteration. Details about the experiments setup are available in appendices C.1 and C.2 respectively. A summary of the results is presented in Table 1. Results show that the new approach proposed for building robots outperforms the *traditional* one in the presence of noise by even an order of magnitude in some cases.

Table 1: Displays the standard deviation ($\sigma$) of the Gaussian noise ($N(0, \sigma)$) applied to each one of the joints in the robot and the corresponding Root Mean Square Error (RMSE) obtained while the robot executes the task when following the *MASA* approach (Figure 1 c) and the *traditional* approach (Figure 1 a). Best result per noise perturbation has been highlighted in **bold**.

| stdev ($\sigma$) [radians] | stdev [degrees] | RMSE for *MASA* approach | RMSE for *traditional* approach |
|---|---|---|---|
| 0,0 | 0 | 0.0015 | **8.34853e-07** |
| 0,01 | 0.573 | **0.0012** | 0.00256271 |
| 0,02 | 1.15 | **0.0054** | 0.0122358 |
| 0,05 | 2.86 | **0.0120** | 0.0131483 |
| 0,1 | 5.73 | **0.0078** | 0.0757341 |
| 0,2 | 11.5 | **0.0372** | 0.0437187 |
| 0,3 | 17.2 | **0.0261** | 0.0444348 |
| 0,5 | 28.6 | **0.0453** | 0.300172 |

Most of the conventional testing conditions are virtually irrelevant when it comes to practical situations, thereby we advocate for adaptable robot mammals. We argue that self-adaptable robots, besides modularity, require not to be programmed but rather trained incrementally using AI techniques. This leads towards more adaptable machines, able to face changes and evolve in parallel with the robotics ecosystem.

---

[3]Brooks refers particularly to mobile robots but we consider that these claims can easily extend to the majority of robots nowadays.

APPENDICES

## A  BIOLOGICAL INSPIRATION

Real life is dominated by performance in sub-optimal conditions. All living creatures are tested in an ever changing environment. All organisms, not only need to be capable to adapt to these changes but also respond to them in an adequate manner as to ensure survival, reproduction and, ultimately, evolutionary success. When it comes to the conception of robots, we advocate for mammals. Self-adaptable creatures that are able to evolve their behaviors to match their environment and morphologies, regardless of the changes suffered. Examples such as dogs that learn to walk missing a leg are common.

Such is the nature.

## B  DESCRIPTION OF THE ROBOT AND TOOLS

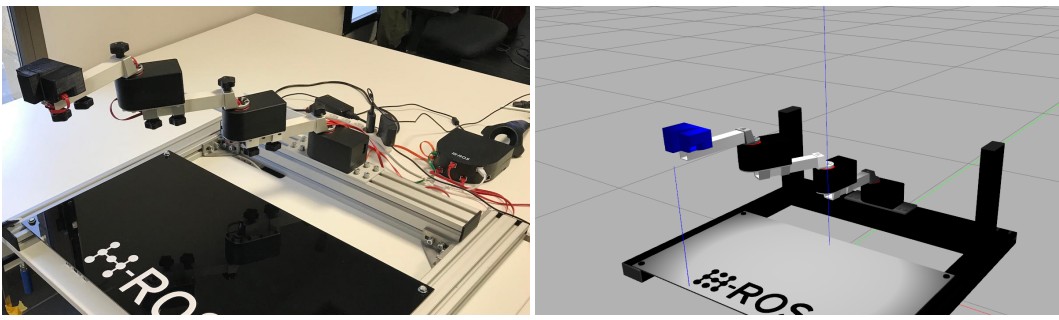

Figure 2: Figure depicts the 3 Degrees-of-Freedom (DoF) real robot in a SCARA configuration (left) and its corresponding simulated model (right).

A 3 Degrees-of-Freedom (DoF) robot in a SCARA configuration is used for the experimental studies. In order to facilitate the configuration and reconfiguration of the robot, modules enabled by the Hardware Robot Operating System (H-ROS) Mayoral et al. (2017) technology have been used. The robot configuration is pictured in Figure 2. A rangefinder has been placed at the last link and is used as a visualization tool for the location of the robot's end-effector. Simulations are performed using Gazebo Koenig & Howard (2004). Both the real robot and the simulated one are programmed using ROS. Accelerated training happened in simulation, using an extension of the work by Kojcev et al. (2018); Zamora et al. (2016), particularized for the SCARA robot described above.

## C  EXPERIMENTS SETUP

### C.1  THE *traditional* APPROACH FOR BUILDING ROBOTS

The *traditional* approach for building robots is depicted in Figure 3. This strategy is better understood as follows:

1. **Buy component**: Refers to the action of acquiring those robot components required to build the robot. Typically, this includes sensors, actuators, communication devices, power mechansims, etc.

2. **Integration of components** (critical section): Many of the devices used in robotics, when compared to each other, typically, consist of incompatible electronic components with different software interfaces. The task of configuring and matching all components in a robot is known as the 'integration effort'. Generally composed by diverse sub-tasks and demanding multidisciplinar knowledge, the integration effort supersedes many other steps in the process of building a robot.

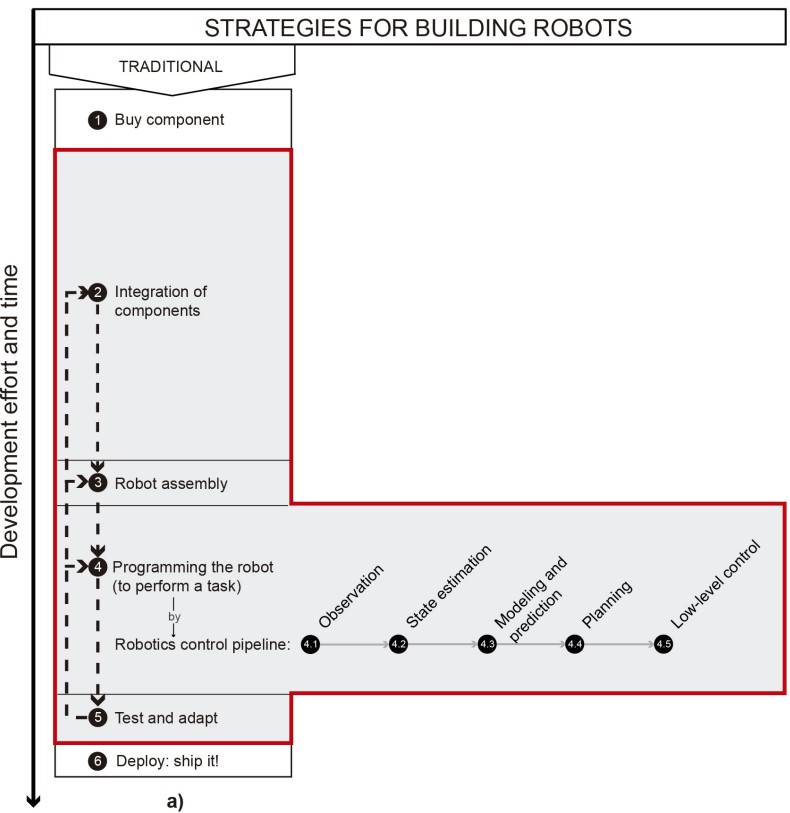

Figure 3: Depicts the *traditional* approach for building robots including a representation of the robotics control pipeline. The critical section is a path that marks a section in the process where each individual change on any of the contained blocks will demand a complete re-execution of all the steps inside the critical section has been highlighted in red.

3. **Robot assembly** (critical section): This step captures the physical construction and assembly of the robot.

4. **Programming the robot** (critical section): Programming the robot is done through the well known *sense-model-plan-act* framework. Figure 3 depicts the robotics control pipeline which captures this framework. In our experiment, we implement the robotics pipeline using ROS as follows:

    - **Observations**: In our robot, the observations correspond with the positions, velocities and efforts of the joint angles. These values are fetched from the servomotor_driver[4] ROS package. The noise introduced and represented in Table 1 has been added to the joint controllers (both in the real robot and in simulation, through modifications in the Gazebo source code).

    - **'State estimation' + 'Modeling and prediction'**: Given the observations from the previous step, the *traditional* approach describes the pose of the robot by inferring a set of characteristics such as its position, orientation or velocity. Considering the SCARA robot, this estimation will be the position of the end-effector calculated from the forward kinematics. In our experiments, we calculate the kinematic data of the robot using several ROS packages. Among them, we made active use of the urdf package[5] –able to read a file which represents the robot model as a tree-like structure– and the kdl package[6] –the default numerical inverse kinematics solver in ROS which

---

[4] servomotor_driver acts as a generic placeholder and does not represent an actual ROS package.
[5] https://github.com/ros/urdf
[6] https://github.com/orocos/orocos_kinematics_dynamics

can be used to publish the joint states and also to calculate the forward and inverse kinematics of the robot–. Mistakes in the observations lead to errors in the state estimation and are typically handled either in the state estimation or in the modeling and prediction step.

- **Planning**: It consists of determining the actions required to execute the task. It is a complex undertaking, specially from a mathematical perspective, since we need to consider limitations in joints, collisions, obstacles, etc. In our experiment, planning implies calculating the points in the space that the end-effector needs to follow so that it can achieve its goal –reach a given point in the space–[7]. We implemented a planning technique using the `moveit` ROS package suite[8]. Particularly, we make use of *move_group* package[9], a node that acts as an integrator of the various abstractions that represent the robot and delivers actions or services depending on the user's needs to facilitate the process of planning. Specifically, in our experiment, *move_group* node collects the joint states and transforms them into actions or services that will be used in the blocks that follow.

  Knowing the starting pose of the robot, the desired goal pose of the robot and the geometrical description of the robot (and the world, both calculated through the `urdf` package and related ones), we proceed to execute motion planning –the technique to find an optimum path that moves the robot gradually from the start pose to the goal pose–. The output of motion planning is a trajectory consisting of joint spaces for each joint in which the links of the robot should never collide with the environment, avoid self-collision (collision between two robot links) and also not violate the joint limits.

- **Low-level control**: The final step in the pipeline consists in transforming the 'plan' into low level control commands that steer the robot actuators to execute a given task. In our implementation, using the `ros_control` ROS package suite[10], after motion planning, the generated trajectory talks to the controllers in the robot using the `ros_controllers`[11] interface. This is an action interface in which an action server runs in the robot, and `move_node` initiates an action client which talks to this server and executes the trajectory on the real robot or on its simulated version.

5. **Test and adapt** (critical section): This step refers to the process of validating the programmed logic in the robot and the overall performance. If valid, the machine can be deployed. Alternatively, one would come back to the integration of components (step 2), the robot assembly (step 3) or the programming of the robot (step 4). A change on a layer will demand a complete re-execution of all the layers below which make the process slow, expensive and time consuming.

## C.2    THE *Modular And Self-Adaptable* (MASA) APPROACH FOR BUILDING ROBOTS

Figure 4 depicts a new approach for building robots that leads to more adaptable and capable machines denoted the *MASA* approach. This strategy for building robots can be summarized as follows:

1. **Buy module**: This step refers to the action of acquiring those robot modules required to build the robot. Since the devices are modules, they are assumed to be interoperable, easy to integrate and re-use (refer to Appendix D for a definition of 'module').

2. **Define task** (critical section): This step refers to the process of defining the goal that the robot should accomplish in a mathematical form so that the learning algorithms can be based on it. Typically, in Reinforcement Learning (RL) –a set of AI techniques– this mathematical expression is captured in what is called a 'reward function' that steers the learning process of the robot.

---

[7]This implies to know all the joint states for each intermediate point in the trajectory that needs to be calculated.

[8]`moveit` is a set of packages and tools for doing mobile manipulation in ROS, providing state of the art software for motion planning, manipulation, 3D perception, kinematics, collision checking, control, and navigation. https://github.com/ros-planning/moveit

[9]https://github.com/ros-planning/moveit/tree/kinetic-devel/moveit_ros/move_group

[10]https://github.com/ros-controls/ros_control

[11]https://github.com/ros-controls/ros_controllers

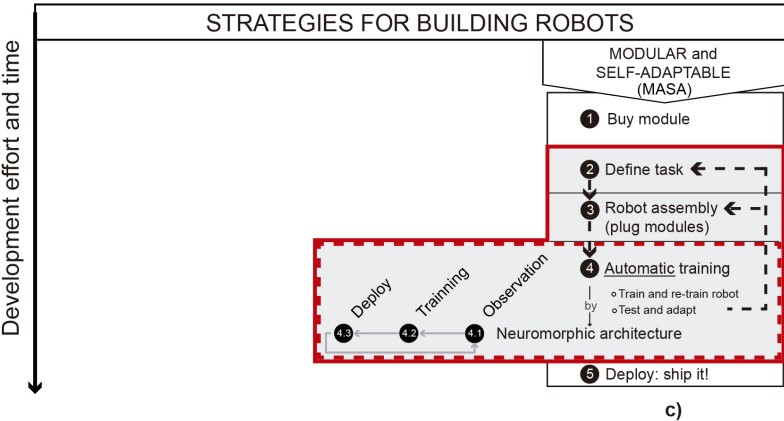

Figure 4: Represents the *Modular And Self-Adaptable* (MASA) approach for building robots. The critical section is underlined in red. Step 4, Automatic training contains a dashed line implying that this process executes automatically and without any human effort.

3. **Robot assembly** (critical section): We capture the physical construction of the robot in this step, which is simplified since all modules interoperate.

4. **Automatic training** (critical section): In this step, together with reconfiguration mechanisms such as those proposed by Mayoral et al. (2017); Zamalloa et al. (2018), we implement AI techniques that allow the robot to continuously integrate the information from its modules and adapt dynamically a neuromorphic model to fit the task defined in previous steps. This way, regardless of the physical changes that happen in the robot (such as additions or removal of modules), the robot will automatically retrain itself for the task.

   In our particular implementation, we extend the Deep Reinforcement Learning (DRL) techniques proposed by Kojcev et al. (2018) and include the reconfiguration ideas cited previously. Specifically, we apply Proximal Policy Optimization (PPO) Schulman et al. (2017), which alternates between sampling data trough interaction with the environment and optimizing the 'surrogate' objective by clipping the policy probability ratio. Noise is introduced on each iteration of the learning algorithm. Figure 5 pictures the learning process of PPO under different noise effects. Figure 6 depicts the result of applying Gaussian noise to the learning process. The figure shows different trajectory outputs, one for each noise perturbation applied.

5. **Deploy**: Once trained, this approach outputs a flag that notifies about the success or failure of the automatic training step. In the case of failure, the user can refine the task definition (step 2) or add additional modules to the robot (step 3) and allow the training process to iterate again automatically (step 4) until success. Figure 7 depicts the resulting poses of the simulated robot trained under different noise perturbations.

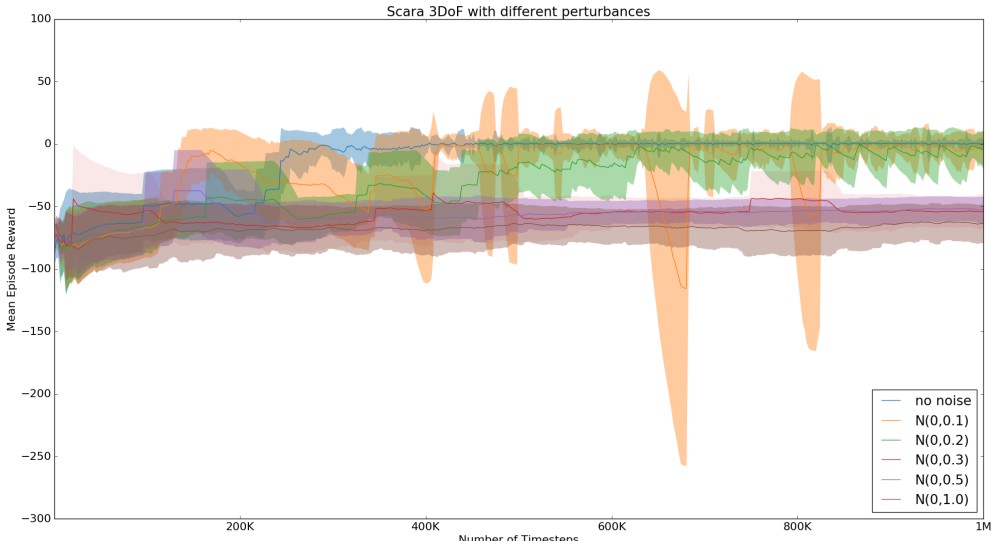

Figure 5: Depicts the learning process of the DRL technique subject to different Gaussian noise effects. This process is part of the *MASA* approach for building robots. Interestingly, when exposed to Gaussian noise perturbations with zero mean and standard deviation of 5.73 and 11.5 degrees per joint, *MASA* produces robots that adapt and whose performance reach similar levels than when no noise is applied.

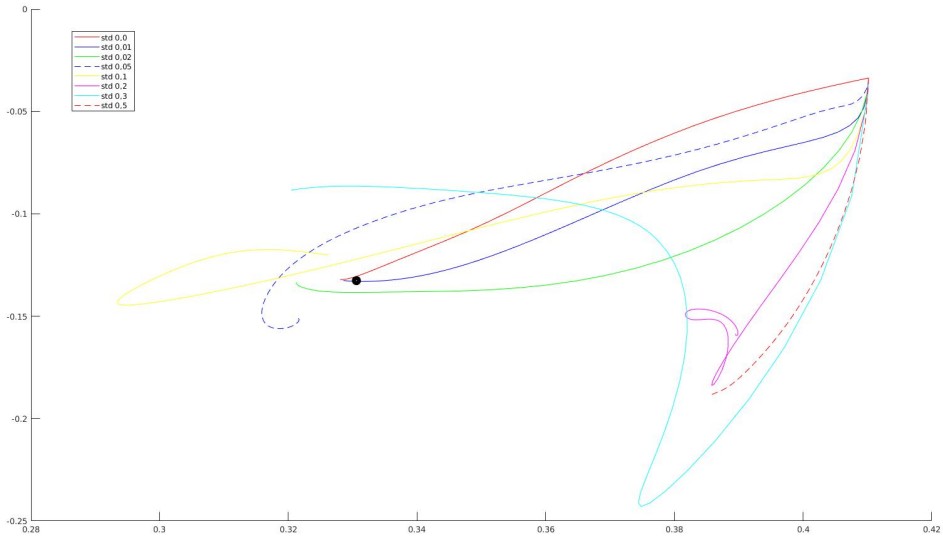

Figure 6: Arbitrary trajectories of each one of the outputs of *MASA* for the given task under different levels of noise. The values in the x and y-axes are in meters (m). An interesting observation is that in the presence of Gaussian noise, *MASA* is able to adapt, and although the trajectory overshoots the target, it returns to the goal improving its accuracy. For example, the yellow line presents the output of *MASA* for a robot where each joint has been subject to an error of zero mean and 0.1 radians (5.73 degrees, refer to Table 1 for more details) of standard deviation. Still, it manages to get to less than 2 centimeters.

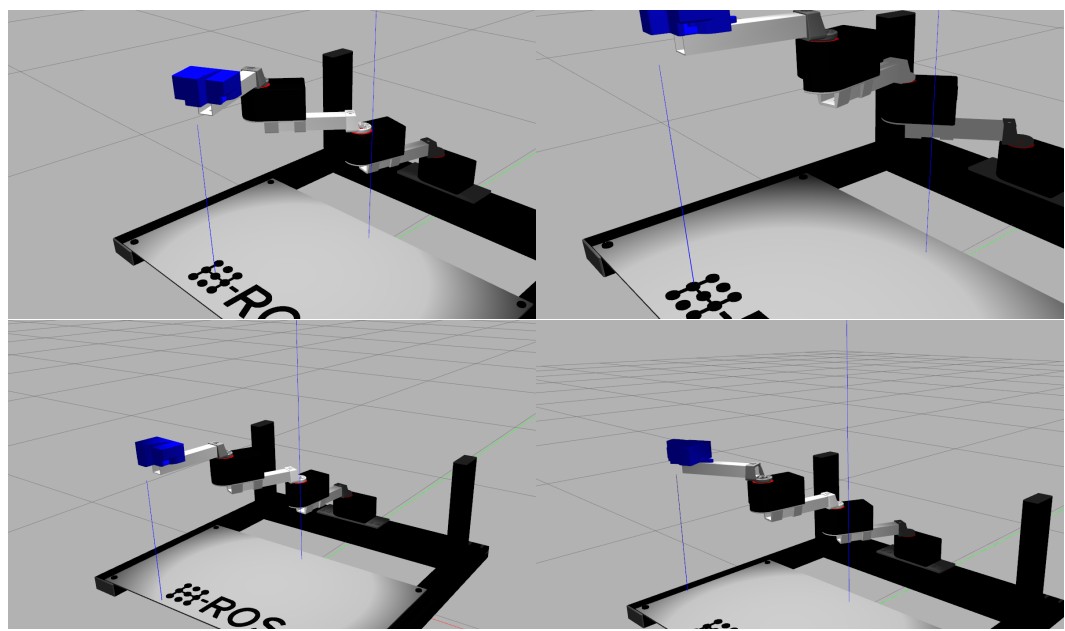

Figure 7: Final positions of the SCARA robot controlled by the neuromorphic technique described in the *MASA* approach for building robots. Each technique has been trained for 1 million iterations using PPO. The figure pictures six different final positions, each corresponding with a robot trained under a different amount of Gaussian noise on its observations: $N(\mu = 0, \sigma = 0)$ (top left), $N(\mu = 0, \sigma = 0.1)$ (top right), $N(\mu = 0, \sigma = 0.5)$ (bottom left) and $N(\mu = 0, \sigma = 1.0)$ (bottom right).

## D  GLOSSARY

The following definitions aim to clarify the language in the context of modular robots:

**Definition 1.** *component a part of something that is discrete and identifiable with respect to combining with other parts to produce something larger (Source: ISO (2017)).*

**Definition 2.** *module component with special characteristics to facilitate system design, integration, interoperability and re-use.*

**Definition 3.** *configuration (noun) the arrangement of a modular robot in terms of the number and type of modules used, the connections between those modules, and the settings for those modules, in order to achieve the desired functionality of the modular robot as a whole.*

**Definition 4.** *modularity: process of designing / building modules to facilitate easy robot configurations for different applications.*

**Definition 5.** *reconfiguration: altering the configuration of a modular robot in order to achieve an intended change in the functionality.*

**Definition 6.** *adaptation: the process of change by which a modular robot becomes better suited to its environment and its goal.*

**Definition 7.** *self-configuration (automatic configuration): achieving configuration of a modular robot through an automated process without human interaction except to initiate the process, if necessary.*

**Definition 8.** *self-reconfiguration (automatic reconfiguration): achieving reconfiguration of a modular robot through an automated process without human interaction except to initiate the process, if necessary.*

**Definition 9.** *self-adaptation (automatic adaptation): achieving adaptation of a modular robot through an automated process without human interaction except to initiate the process, if necessary.*

### ACKNOWLEDGMENTS

Endika Gil challenged our ideas and contributed with insights about the underlying biological relationships between the robot methods we treat and real organisms. Laura Alzola Kirschgens con-

tributed reviewing our content, making our text easier to read and more understandable. Iñigo Muguruza, Lander Usategui, Asier Bilbao, Carlos San Vicente and Jorge Lamperez helped setting up the robot experiments and provided support in the validation of *MASA*, our new robot building process. Carlos Uraga, Aitor Rioja and Patxi Mayoral supported this work through conversations with manufacturers.

This paper describes research done at Erle Robotics S.L. owned by Acutronic Link Robotics AG. Support for the research is provided in part by the business development basque agency (SPRI) through the HAZITEK 2017 program.

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
