# OpenReview forum: "Towards self-adaptable robots: from programming to training machines"
_ICLR.cc/2018/Workshop — Reject_

### Official Review · AnonReviewer3 · 2018-02-28
**Interesting, but quite off-topic for ICLR and insufficient knowledge of the literature**

**Rating:** 3
**Confidence:** 4

**Review:**

As the last sentence of the abstract says, this paper advocates for mammals...
No, I'm joking.

This paper advocates for a non-standard approach to robot design based on connecting general purpose modules and using deep reinforcement learning to let the robot learn how to acheive goals. An experiment is performed which shows that the proposed Modular And Self-Adaptable (MASA) approach is more robust to sensor noise than the traditional approach.

There are two main issues.

First, the focus of the paper is on the robotics design methodology, which is quite off-topic in a conference about learning representations. The work should rather have been focused on the experiments performed in appendix C2 where PPO (Schulman 2017) is used to learn a task with the modular robot, but unfortunately this is not the case.

Second, the authors build on Brooks' 86 vision of bottom-up robotics, but they seem to ignore a lot of more recent research about modular and self-adaptive robotics, such as the work of Josh Bongard, Hod Lipson and many other very visible researchers in the domain.

To me, these two issues are enough to reject the paper.

Apart from that, though I'm quite sympathetic to the overall methodology, my feeling is that the performed experiment is not strong enough to support it, and that the very short "Biological inspiration" appendix is detrimental to the quality of the paper rather than it improves it.

---

### Official Review · AnonReviewer1 · 2018-03-10
**What's the difference between MASA and applying ML methods to robotics?**

**Rating:** 3
**Confidence:** 4

**Review:**

The whole point of this paper appears to be that if learning - called "automatic training" in the manuscript - is used to control a robot, it is possible to reduce development effort/time (compared to traditional control approaches). Isn't this one of the main reasons to use learning-based approaches/ML in the first place? What is new about this paper?

I fail to see the novelty in this work. You compare an existing learning-based method (PPO) to classic control (kinematics + some ROS packages for planning/low-level control) on a very simple simulated robot (planar arm).
What are the sensors/feedback used for closed-loop control? Actuator positions?

---

### Decision · Program_Chairs · 2018-03-20
**ICLR 2018 Workshop Acceptance Decision**

**Decision:**

Reject

**Comment:**

Based on the reviews, this paper has not been accepted for presentation at the ICLR workshop. However, the conversation and updates can continue to appear here on OpenReview.